# Advances in Magnetoresistive Biosensors

**DOI:** 10.3390/mi11010034

**Published:** 2019-12-26

**Authors:** Diqing Su, Kai Wu, Renata Saha, Chaoyi Peng, Jian-Ping Wang

**Affiliations:** 1Department of Chemical Engineering and Material Science, University of Minnesota, Minneapolis, MN 55455, USA; suxxx435@umn.edu; 2Department of Electrical and Computer Engineering, University of Minnesota, Minneapolis, MN 55455, USA; wuxx0803@umn.edu (K.W.); saha0072@umn.edu (R.S.); peng0288@umn.edu (C.P.)

**Keywords:** magnetoresistance, biosensors, MNPs, immunoassay, genotyping, flexible devices, brain mapping

## Abstract

Magnetoresistance (MR) based biosensors are considered promising candidates for the detection of magnetic nanoparticles (MNPs) as biomarkers and the biomagnetic fields. MR biosensors have been widely used in the detection of proteins, DNAs, as well as the mapping of cardiovascular and brain signals. In this review, we firstly introduce three different MR devices from the fundamental perspectives, followed by the fabrication and surface modification of the MR sensors. The sensitivity of the MR sensors can be improved by optimizing the sensing geometry, engineering the magnetic bioassays on the sensor surface, and integrating the sensors with magnetic flux concentrators and microfluidic channels. Different kinds of MR-based bioassays are also introduced. Subsequently, the research on MR biosensors for the detection of protein biomarkers and genotyping is reviewed. As a more recent application, brain mapping based on MR sensors is summarized in a separate section with the discussion of both the potential benefits and challenges in this new field. Finally, the integration of MR biosensors with flexible substrates is reviewed, with the emphasis on the fabrication techniques to obtain highly shapeable devices while maintaining comparable performance to their rigid counterparts.

## 1. Introduction

Magnetoresistance (MR) effect refers to the change of resistance under the influence of an external magnetic field. The capability of converting magnetic signals to electrical signals has led to the successful implementation of sensors based on the MR effect. To date, MR sensors have been widely employed in various applications such as hard disk drives [1], magnetic random access memory (MRAM) [2], pressure measurement [3,4], current sensing [5,6], position sensing [7,8,9], and biosensing [10,11,12] (Figure 1). The first application of MR sensors to detect biological signals was proposed about 20 years ago by Baselt et al. [13], and they have attracted great attention since then [14,15,16]. Compared to other traditional biosensors such as optical-based sensors, MR biosensors exhibit lower background noise and are less affected by the environment of the biological sample, such as pH and temperature [17]. An extremely high sensitivity for a large area GMR sensor was demonstrated with the capability of detecting 150 magnetic nanoparticles (MNPs) per sensor [17]. Current nanofabrication technologies also enable the integration of MR sensor array onto one single chip, which facilitates the multiplexed detection of biomarkers [18] as well as the mapping of biomagnetic field with high spatial resolution [19]. Moreover, the miniaturization of the MR biosensor chips makes it possible to incorporate them into point-of-care (POC) devices for rapid and onsite detection [20,21].

Herein, we will first introduce the basics of several types of MR effects including anisotropic magnetoresistance (AMR), giant magnetoresistance (GMR), and tunneling magnetoresistance (TMR). In the following section, the fabrication techniques will be reviewed in several aspects. We will include the design of both the geometry of the sensing area and the material/thickness of the passivation layer. The integration with magnetic flux concentrators (MFCs) and the microfluidic channels will also be discussed as they are widely employed in MR biosensors for increased sensitivity and efficient detection of biofluidic samples. Subsequently, the surface functionalization of sensor surfaces with different kinds of bioassays including sandwich assays, competitive binding assays, direct assays, and aptamer-based assays will be introduced in the same section. In Section 4 and Section 5, the recent progress in both the detection of protein biomarkers and the genotyping process with MR sensors will be reviewed. Besides biological detection based on magnetic tags, the direct detection of biomagnetic field will be summarized in Section 6, with an emphasis on brain mapping technologies. Finally, the integration of MR sensors with flexible substrates by direct deposition, substrate transfer, and the printing of MR flakes will be discussed. It is expected that a comprehensive and up-to-date review of the MR biosensing technologies can be achieved. The authors’ perspectives on the pros and cons of different MR biosensors and their future development for next-generation magnetic biosensing will also be provided.

## 2. Magnetoresistive (MR) Devices

### 2.1. Anisotropic Magnetoresistance (AMR)

The discovery of AMR was made by William Thomson which can be dated back to the year of 1857 [24]. It was found that the resistivity for both Ni and Fe increased when the current was parallel with the magnetization direction and decreased when the current was perpendicular to the magnetization direction. As a quantitative description of the AMR effect, the AMR ratio was introduced and can be defined as $\Delta \rho /\rho_{av}$, with $\rho_{av}=\frac{1}{3}\rho_{∥}+\frac{2}{3}\rho_{⊥}$ and $\Delta \rho =\rho_{∥}-\rho_{⊥}$. Here, $\rho_{av}$ is the average resistivity, Δρ is the anisotropic magnetoresistivity, $\rho_{∥}$ is the resistivity with current parallel to the magnetization, and $\rho_{⊥}$ is the resistivity with the current perpendicular to the magnetization [25]. During the following 100 years, much attention had been attracted to this phenomenon and its physical origin [26,27,28]. In 1936, Mott firstly brought up a two-current model suggesting that the transport properties of the ferromagnetic materials can be explained by expressing the total conductivity as a sum of the conduction in spin up and spin down electrons connected in parallel [29]. Since in Ni, Co, Fe, and their alloys, the stronger s-d scattering only exists for spin down electrons, thus the resistivity will be higher in spin down channels. This anisotropic scattering process induced by the spin-orbit interaction is the origin of the AMR effect. The model was subsequently demonstrated both experimentally and quantitatively by Fert and Campbell [26]. Despite this groundbreaking work in the field of magnetoresistance, the resistance change at room temperature is only 2%, which makes it hard to build up AMR-based devices in most of the applications until the discovery of giant magnetoresistance (GMR). A detailed review of the AMR effect and the experimental results on thin films and bulk materials can be found in Ref. [25].

### 2.2. Giant Magnetoresistance (GMR)

In 1988, Baibich et al. observed a two-fold resistance decrease in the (001)Fe/(001)Cr superlattices grown by molecular beam epitaxy (MBE) under a magnetic field of 2 T and temperature of 4.2 K [30]. A similar effect was also observed later in the Fe-Cr-Fe system by Binasch and Grünberg [31]. This resistance change is significantly higher than the AMR effect, and is thus named as giant magnetoresistance. GMR effect exists in metallic structures with alternating ferromagnetic and nonmagnetic layers. Under an applied magnetic field, the magnetization directions of two adjacent ferromagnetic layers can be either parallel or antiparallel depending on the orientation of the external field, which corresponds to low- or high-resistance states, respectively. A breakthrough towards the industrial application of the GMR devices was made by Parkin et al., who demonstrated the first Co/Cr and Co/Ru GMR multilayer structures through magnetron sputtering techniques [32]. Since then, many efforts have been made towards the commercialized application of GMR-based devices, such as biosensors [13,21,33], position sensors [7,34], and magnetic random access memory (MRAM) [35,36,37]. An example of the GMR stack structure is shown in Figure 2a.

Although the GMR effect was at first discovered and mostly investigated in thin film stacks, it can also occur in other systems without the traditional layer structures. In 1992, it was demonstrated by Xiao et al. that GMR can be measured in magnetically inhomogeneous media [38]. Phase-separated Co-Cu and Fe-Cu samples were prepared by dc magnetron sputtering with Co and Fe particles embedded in Cu matrix. A GMR ratio of 13% at 5 K was observed for Co_38_Cu_62_ after annealing at 480 °C. Similarly, a GMR ratio of 9% was observed in the Fe_30_Cu_70_ system. Other material systems such as Co-Au [39,40], Co-Ag [41,42], and Fe-Ag [43] granular films were also investigated later on. Since the granular GMR effect largely depends on the spin-dependent interfacial electron scattering and the inter-particle coupling, multiple factors such as particle size, inter-particle distance, annealing temperature, and ferromagnetic volume fraction [42,44,45,46] need to be considered in the design of the granular GMR systems. To obtain better control over the size and the volume fraction of the magnetic particles, bottom-up approaches, where the magnetic particles were pre-synthesized and then embedded into the nonmagnetic matrix, were also demonstrated [47,48,49]. Both granular and thin film-based GMR structures have been employed in biosensing applications, which will be reviewed in subsequent sections.

### 2.3. Tunneling Magnetoresistance (TMR)

The tunneling magnetoresistance was originally found by Michel Julliere in a Fe-Ge-Co multilayer in 1975 [50]. The magnetic tunnel junction (MTJ) has an essential sandwich structure that two ferromagnetic layers are separated by an insulator layer. As shown in Figure 3a, the ferromagnetic materials have a split band structure due to the spin polarization. In the parallel magnetization state, the majority spins in the tunnel through the barrier and enter the majority band and so do the minority spins. By implementing a two-current model as shown in Figure 3b, it will result in a low-resistance state *R* = 2*R_p_R_ap_*/(*R_p_* + *R_ap_*). In the antiparallel state, the majority spins could only tunnel through the barrier and enter the minority band, and the minority spins tunnel and enter the majority spins. In this case, it will result in a high resistance *R* = (*R_p_* + *R_ap_*)/2.

A typical MTJ structure is shown in Figure 2b. When the magnetizations of two ferromagnetic layers are parallel, the majority spins get across the insulator layer and enter the majority bands. The minority spins enter the minority bands, causing a low-resistance state (*R_P_*). When the magnetizations turn to an anti-parallel state, the majority spins could only enter the minority bands and minority spins enter the majority bands. Because there are fewer spins to receive tunneling, it results in a high-resistance state (*R_AP_*). The tunneling magnetoresistance ratio is defined as:$$TMR=\;\frac{R_{AP}-R_{P}}{R_{P}}$$

A typical transfer curve of the magnetoresistive sensors including MTJs can be found in Figure 2c. There are two important parameters that can be obtained from the transfer curve, i.e., MR ratio and sensitivity. While the MR ratio is characterized by the change of resistance when the magnetic field sweeps from one direction to the other, the sensitivities of the MR devices are characterized by the slope of the transfer curve at a magnetic field strength of interest. It is worth mentioning that there is a tradeoff between the sensitivity and linearity for both GMR and TMR sensors. A large linear regime can be easily achieved in GMR sensors, which often comes with the sacrifice of the sensitivity. While TMR sensors possess high sensitivity, their linear region is relatively narrow. Another important parameter of magnetoresistive sensors we need to mention here is the signal-to-noise ratios (SNRs). TMR sensors would only have better performances if the SNRs are higher than GMR sensors. Although most of the TMR sensors possess a higher MR ratio compared to GMR sensors, the shot noise arising from the discontinuities in the conduction medium such as tunnel barrier can result in increased noise level [53].

There has been much effort to increase the MR ratio of MTJs since their first discovery. The TMR ratio which Julliere found is only 14% at the temperature of 4.2 K. In the following twenty years, the MTJ research study developed very slowly due to the limited equipment and film-deposition technology. Later after the observation of the GMR effect, in 1995, Miyazaki [54] from Japan and Moodera [55] from MIT observed a TMR ratio of 18% and 12% in amorphous Al_2_O_3_-barrier MTJ, respectively. Following the theoretical calculation [56], a high TMR ratio was found in MgO-barrier MTJs. In 2004, Parkin et al. [57] reported a TMR ratio of 220% at room temperature and Yuasa et al. [52] found a TMR ratio of 180% at room temperature.

## 3. Magnetoresistive Biosensors: From Fabrication to Surface Modification

### 3.1. Fabrication

#### 3.1.1. Design of Sensing Elements

One of the ultimate goals in the design of the sensing elements is to achieve the highest sensitivity, which requires a comprehensive understanding of the interaction between the sensing elements and the magnetic tags. Wang et al. investigated the sensor signal dependence on the vertical position of a 16 nm MNP [33]. It was shown that the sensor signal did not decrease monotonically with increased distance from the MNP to the sensor surface as one would expect. Instead, the maximum sensor signal occurred at a certain distance of 70 nm. This was attributed to the fact that the direction of the stray field from the free layer of the MR sensor is opposite to the direction of the external field, thus reducing the field from the MNP in proximity to the surface. Klein et al. explored different designs of sensor configurations where the MNPs were either exposed to or isolated from the free layer stray field from the edge of the GMR sensors [58]. With both theoretical and experimental evaluations, it was found that no gap fillings or thick passivation layers are needed around the edge of the sensing area to eliminate the contribution from the MNPs influenced by the stray field from the free and fixed layers. Instead, the MNPs on the edge of the sensor interferes constructively with the MNPs on top of the sensor. More sensing configurations were also explored later by Lee et al. [59], which also suggested that the MNPs on the edge of the sensor could contribute more sensor signals. However, the higher sensor signal from the edge of the sensing element can also lead to the nonuniformity of the sensor signal, which means the effective sensing area should still be near the center of the sensor for large-aspect-ratio spin valve sensors [60,61].

In addition to the improvement of the sensitivity and the signal uniformity, the throughput of the measurement is another important parameter in the design of the sensing devices, especially for point-of-care (POC) applications. The time needed for the measurement depends on various processes, such as the binding affinity of the target analyte to the capture antibodies or DNAs, the efficiency of the washing process, and the readout time of the electrical signal from the sensing circuit. There have been multiple approaches to reduce both the assay preparation time and the measurement time such as the integration of microfluidic channels, optimization of the magnetic assays, as well as the improvement of the sensing circuit. While the assay preparation time can be as long as several hours or even several days, the functionalization of the sensing chips can be performed in advance in the laboratories. The onsite measurement for the POC devices only includes the time for the binding of the magnetic tags, which can be performed in less than 10 min [20,21].

#### 3.1.2. Design of Passivation Layer

Although the nanofabrication processes of MR biosensors are almost identical to the other types of MR sensors, the need for direct contact with biological tissues or aqueous solutions containing biological samples makes the passivation of the sensor surface a crucial step during the design of the sensing platforms. There are three functions for the passivation layers: (a) isolate the magnetic materials from the biological environment since most of them are highly cytotoxic; (b) isolate the electrical circuits from the aqueous solution; (c) provide the basis for the subsequent sensor surface biofunctionalization. However, as the magnetic signal from the magnetic tag decays rapidly with respect to their distance to the sensor surface, the thickness of the passivation layer needs to be optimized to achieve the aforementioned goals while maintaining high sensitivity. The deposition of the passivation layers for these MR sensors must be toxic chemicals free, meaning chemical vapor deposition (CVD) techniques must be avoided. In addition, the passivation layers must be of low energy states such that it encourages surface immobilization of the biological molecules. The most commonly used passivation layers for MR sensors are SiO_2_, Al_2_O_3_, and Si_3_N_4_, with a thickness of 50 nm to 200 nm [21,62,63,64]. One example of surface functionalization based on the passivation layers is as follows: ultraviolet oxygen plasma (UVO) treatment will firstly be employed to remove any contaminants on the surface of the sensor. Subsequently, amino groups will be bound to the sensor surface through the reaction between the hydroxyl groups of SiO_2_ and chemicals such as 3-aminopropyltriethoxy silane (APTES) and 1-ethyl-3-(3-dimethyl aminopropyl) carbodiimide hydrochloride. The aldehyde groups will then be introduced, which can bind to the amino groups on the antibodies to be functionalized on the sensor surface [18,65]. Similar biochemical process can also be employed in the functionalization of oligonucleotides due to the amino groups in their structure. Other techniques such as the functionalization of DNAs or antibodies on gold surfaces through thiolated probe immobilization are also frequently employed [66]. The cytotoxicity of the passivation layers on the MR sensors has been evaluated by Moretti et al. [67], where a three-layer capping of physical SiO_2_/Si_3_N_4_/SiO_2_ was deposited on the surface of an MTJ sensor by physical vapor deposition (PVD). It was found that the neuron cells can be grown on the sensor surface for 20 days while maintaining their functionality. The performance of MTJ sensors also remained unchanged for several weeks, proving that the passivation layers were fully biocompatible.

#### 3.1.3. Integration with Magnetic Flux Concentrator (MFC)

During the development of MR biosensors, the enhancement of the sensitivity has become one of the most crucial goals due to the need of biomarker detection with ultra-low concentrations as well as the direct detection of weak biomagnetic field from human organs such as cardiac or neural signals, which requires sensitivities of sub-pT or even fT level. The integration of MR sensors with magnetic flux concentrators (MFCs) has been a hot topic over the past 10 years. There are several types of MFCs. In superconducting MFCs, a supercurrent is generated by the magnetic field applied to a superconducting loop. Since part of the loop is narrowed in geometry, the local current density is increased, which leads to the amplification of the localized magnetic field [68]. As an example, hybrid sensors composed of GMR elements and a niobium loop were fabricated and were used to detect magnetocardiography (MCG) signals, which exhibited comparable sensitivity to Superconducting Quantum Interference Devices (SQUIDs) [19]. However, due to the involvement of superconducting materials, the working temperature for these devices is very low, which increases the complexity of the setup of the sensing platform. The other kind of MFCs is based on soft-ferromagnetic materials [69]. By concentrating the magnetic flux into the senor region, these MFCs are capable of increasing the sensitivity by hundreds of times while working at room temperature [70]. It was shown that by integrating MTJs with a CoCrPt permanent magnet and a CoZrNb MFC, the lowest detectable field of 49 pT/Hz^0.5^ was achieved [71] (Figure 4). More recently, MTJs integrated with double staged MFCs were also reported. By incorporating two MFCs, one on the MTJ chip level and the other on a more macroscopic level, a detectable field limit of 30 pT/Hz^0.5^ was achieved with a low coercivity of 0.12 Oe. Besides MTJs, highly compact spin valve (SV) sensors have also been fabricated and integrated with CoZrNb MFC. The sensitivity was increased from 0.18%/mT to 0.37%/mT with a gain of 20.7 [72].

#### 3.1.4. Integration with Microfluidic Channels

With the growing demand for lab-on-a-chip platforms, the integration of MR sensors with microfluidic channels has become a necessity for the detection of biomarkers with a small volume of biological fluids. Among all the materials for microfluidic channels, polymers stand out for their transparency, biocompatibility, flexibility, and cost-effectiveness. The fabrication processes of a microfluidic channel usually include a photolithography process where a photoresist mold with the desired channel configuration is made, followed by pouring the channel materials (such as PDMS and SU-8) into the mold, curing of channel materials, and removing the mold by corresponding developers [73,74,75]. Recently, PDMS microfluidic channels made from 3D printed molds were also proposed [74]. While the fabrication of microfluidic channels is relatively straight-forward, the integration of the polymeric channels with the silicon-based MR sensor chips is challenging. Physical bonding of the polymer channels can be realized by applying pressure over the channel [76]. Alternatively, the silicon substrates and the microfluidic channels can be treated with oxygen plasma to form irreversible chemical bonds [77]. Besides the integration techniques, there have also been extensive studies on the behaviors of MNPs in the microfluidic channels as well as the manipulation of MNPs through various configurations of patterned magnetic materials and different shapes of microfluidic channels [78,79,80]. One example of the manipulation of MNPs in microfluidic channels is through the controlled nucleation, displacement, and annihilation of the magnetic domain walls. It has been shown that the MNPs in the channel can be captured by the stray field from the domain walls, and thus can be manipulated along the desired path with the precision of 100 nm by altering the shape of the domain wall conduit [81,82]. By integrating with AMR sensors, the real-time detection of the transit of a single magnetic bead during the manipulation has also been realized [83].

### 3.2. Device Surface Chemical Modification

#### 3.2.1. Sandwich Assay

Sandwich assay is one type of robust and sensitive immunoassay method for measuring the presence or concentrations of antigen (or any target analytes of interest) from liquid samples. The antigens are quantified through the specific binding between two layers of antibodies: the capture and the detection antibodies. These two layers of antibodies can either be monoclonal or affinity-purified polyclonal antibodies, but they must bind to non-overlapping epitopes on the antigen.

The general protocol of a sandwich assay is as follows (see Figure 5) [20,64,84,85,86,87]:Immobilize capture antibodies onto sensor surface (e.g., AMR, GMR, and TMR sensor surface, etc.). Wash away unbound capture antibodies.The surface is then coated with blocking buffer (i.e., bovine serum albumin (BSA)) to block the nonspecific binding sites. Wash away blockers.Add a biological sample of antigens to the surface. The antigen is captured and immobilized onto the surface through specific binding to capture antibodies. Wash away unbound chemicals (including unbound antigens).Biotinylated detection antibodies (usually diluted in blocking buffer) are added to the surface for binding to the antigens. Wash away unbound biotinylated detection antibodies. After this step, only the capture antibody-antigen-biotinylated detection antibody complexes are left on the surface.Streptavidin-coated MNPs are added to the surface. The streptavidin binds specifically to the biotin from the detection antibody.An externally applied magnetic field, combined with the magnetic properties of MR sensors, is applied to convert the magnetic stray field from MNP into an electrical signal. The resistance or voltage or magnetoresistance of the MR sensor is measured to determine the presence and quantity of antigens.

#### 3.2.2. Competitive Binding Assay

Competitive binding assay is based on the competition of labeled and unlabeled antigens (or antibodies) for a limited number of antibody (or antigen) binding sites [88]. As shown in Figure 6a, in a competitive binding assay with capture antibodies coated on the MR sensors, the MNP-labeled antigens are competing with the unlabeled antigens from testing samples. In this case, a higher electrical/magnetic signal indicates a lower concentration of antigens from the testing sample. The steps for the capture antibody coated sensing are as follows [89]:Capture antibodies are immobilized on the MR sensor surface. The surface is coated with blocking buffer to block the nonspecific binding sites.Add both MNP-linked antigens (labeled antigens) and antigen-containing sample (unlabeled antigens) to the sensor surface. Cumulative competition occurs between the two antigens for the antibody binding sites.Wash away unbound antigens. The MR sensor converts the number of MNPs into an electrical/magnetic signal. The signal strength of MR sensor determines the presence and quantity of target analytes.

Similarly, for the competitive binding assays with antigens coated on the MR sensors, the MNP-labeled antibodies are competing with the unlabeled antibodies from the testing sample (as shown in Figure 6b). As a result, a higher electrical/magnetic signal indicates a lower concentration of antibodies from the testing sample.

#### 3.2.3. Direct Assay

Compared to the sandwich and competitive binding assays, the direct assay [90,91,92] is less common in biosensing areas. The steps of MR-based direct assay follow the mechanism below:The testing sample containing target antigens (or antibodies) is added to the MR sensor surface, where it is given time to adhere to the sensor surface. Unbound antigens (or antibodies) are washed away followed by the blocking buffer to block the nonspecific binding sites.MNP-linked antibodies (or MNP-linked antigens) are added, which binds specifically to the testing antigens (or antibodies) that immobilized on the sensor surface.Wash away unbound complexes. The MR sensor converts the number of MNPs into electrical/magnetic signals. In this case, a stronger signal indicates a higher concentration of the target antigens (or antibodies) from the testing sample.

As shown in Figure 7a, the target antibodies from testing samples are immobilized on the MR sensor surface, followed by adding MNP-linked antigens which specifically bind to target antibodies. Vice versa, as shown in Figure 7b, the antigens and antibodies switch their position. In most MR biosensing systems, the positive control sensors are usually covered with biotin and the streptavidin-coated MNPs directly bind to biotins, which yields very high electrical/magnetic signals from MR sensors. This biotin-streptavidin-coated MNP structure is one of the most commonly used direct assays (see Figure 7c).

#### 3.2.4. DNA-Based Assay

The foregoing assay methods are mostly based on the specific binding between antigen and antibody. Another very popular assay method is based on DNAs, which has been actively applied in the genotyping detections. As shown in Figure 8, the general protocol of the aptamer-based assay is as follows [63,93,94,95,96]:Immobilize probe DNA onto MR sensor surface. Wash away unbound probe DNA.Apply testing sample containing biotinylated target DNA strands to the MR sensor surface and incubate to allow the complementary target DNA to hybridize with the probe DNA.Wash away unbound target DNAs and noncomplementary DNAs.Streptavidin-coated MNPs are added to the MR sensor surface and captured by the remaining target DNAs through the biotin-streptavidin conjugation. The MR sensor converts the number of MNPs into electrical/magnetic signals.

## 4. Magnetoresistive Biosensors for Immunoassay Applications

### 4.1. AMR-Based Biosensors

In 2002, the first AMR biosensor prototype was proposed by Miller et al. [97], where a ring-shaped AMR sensor made of NiFe was employed to detect the radial fringing field from a NiFe microbead. In the absence of a fringing field, the magnetization in the device is circumferential, resulting in maximum magnetization under the application of an electrical current. If a radial magnetic field is present, the magnetizations in the device rotate toward the radial direction, leading to a minimized AMR state since the electrical current within the device is now perpendicular to the magnetizations. It was found that the AMR response was peaked when the microbead was directly above the center of the ring and then rapidly decreased to zero when the microbead moved outside the ring, which made this device a possible model for the detection of magnetic tags for biomedical applications. Later on, much effort has been made to improve the performance of AMR sensors by incorporating an exchange bias layer [98], a multilayer stack structure [99], a barber pole configuration [100], etc. Recently, AMR sensors based on flexible polymeric substrates were fabricated [101]. With a barber pole configuration, a linear voltage output as a function of the applied field was observed without any bias condition. It was found that the performance of the device can be enhanced by reducing the surface roughness of the film after the coating of a buffer layer. The optimal limit of detection (LOD) of the fabricated sensors was 150 nT.

### 4.2. GMR-Based Biosensors

GMR biosensors are the most studied type of MR biosensors. Two different layer structures have been employed in biosensing: the multilayer system and the spin valve system. Multilayer GMR stacks are composed of alternating nonferromagnetic materials and ferromagnetic materials. Due to interlayer coupling and stray field coupling, the adjacent ferromagnetic layers tend to align in antiparallel configurations [102]. The sensitivity of multilayer GMR structures was found to increase with increasing ferromagnetic layer thickness, while the GMR ratio decreases with increasing ferromagnetic layer thickness [103]. Unlike multilayer structures, the spin valve system only consists of two ferromagnetic layers separated by a non-ferromagnetic layer. The magnetization of one ferromagnetic layer is fixed, while the magnetization of the other ferromagnetic layer can rotate freely. Linear response curves can be achieved in spin valve sensors by the perpendicular configuration of magnetizations through the manipulation of shape anisotropy or a bias field. Although spin valve sensors exhibit better linearity and can work at lower magnetic fields compared to multilayer sensors, they usually possess lower MR ratios since they have only two ferromagnetic layers contributing to the GMR effect.

Since the first proposal of a GMR biosensing system (bead array counter) by Baselt et al. [13], both the multilayer and the spin valve structures have been widely applied for immunoassays. Although the sensing scheme was reported in 1998, the first GMR-based immunoassay was carried out in 2003 by employing spin valve sensors to detect a single MNP via biotin-streptavidin interaction [104]. Later on, the GMR-based biosensing system has been tested for the detection of various biomarkers such as prostate cancer [105], ovarian cancer [18], lung cancer [17], etc. The detection of other types of analytes, such as viruses [21,64,86], bacterias [106,107], and toxic ions [93] have also been demonstrated. Besides single-analyte detection, the possibility of multiplexed immunoassay was also explored. By functionalizing different capture antibodies on different GMR sensors, the sensor signal corresponding to different analytes can be readout simultaneously. Osterfeld et al. succeeded in multiplexed detection of seven different biomarkers with fM sensitivity [62]. Wang et al. developed a multiplexed probe station system that was capable of detecting three biomarkers at the same time from unprocessed human serum with comparable sensitivity to that from buffer solution [85]. Recently, multiplexed detection has also been realized in portable GMR biosensing systems, with the capability of detecting up to 12 kinds of tumor biomarkers [18,108]. In order to detect multiple biomarkers, the packing density of the sensors needs to be increased, which requires more complexed circuit design. Furthermore, the readout time for the measurement will also increase with the number of sensors. The optimization of the microfabrication processes as well as the circuit board is needed to address these challenges for POC applications. To date, the signal readout time for one sensor can be reduced down to 1 s, which makes the total readout time less than 1 min for 50 sensors [64]. Although there is a tradeoff between the number of target biomarkers and the complexity of the circuit design, the magnetoresistive sensor-based POC devices with the ability to perform multiplexed detection are still promising candidates for diagnostic applications as the technical difficulties from the increased number of sensing elements can be easily addressed with the current development of nanotechnologies.

Apart from the improvement in the magnetic immunoassays, the magnetic biomarkers, sensor configuration, and the sensing circuit have also been widely studied to drive the MR sensing system toward POC applications. Several handheld systems have been successfully developed over the past few years with moderate sensitivities for biomarker detections both in buffer solution and biological media [20,21,64] (Figure 9). Meanwhile, some fundamental studies on the interaction between the sensing elements and the magnetic tags were conducted to optimize the configurations of the GMR sensors. It was reported that the sensor signal level is dependent on the position of magnetic tags on the sensor surface. The viability of utilizing large-area sensors based on reverse nucleation mechanism to replace the most commonly used high-aspect-ratio stripes was also demonstrated [109,110,111]. Simulation results have shown that a reverse nucleation mechanism occurs during the switching of large-area GMR sensors, which leads to more localized switching behavior compared to the traditional stripe-shaped sensors. It was reported that with the potential to exhibit more uniform signal changes with respect to different MNP locations on the sensor surface, large-area GMR sensors become promising candidates for the detection of ultra-low concentration biomarkers with high reliability. MNPs with high magnetic moments can contribute higher stray field per particle, thus increasing the sensitivity of the MR biosensors [112,113]. It was found that by employing FeN MNPs with high magnetic moments, as few as 150 MNPs can be detected with large-area GMR biosensors, leading to ultra-low sensitivity in biological detection [17].

### 4.3. TMR-Based Biosensors

Comparing to GMR-based sensors, MTJ sensors offer a higher MR ratio and higher sensitivity. However, the top electrodes of MTJ sensors are often thicker than GMR sensors, which makes the sensing field of the free layer smaller and thus limits the application. The high intrinsic noise of MTJ sensors is another challenge for achieving ultra-high sensitivities. Lei et al. [114] reported liver cancer detection using an MgO-based MTJ sensor by detecting the MNP labeled alpha-fetoprotein (AFP). The MgO-based MTJ sensor was bio-functionalized with AFP capture antibodies and MNPs were functionalized with AFP detection antibodies. A sandwich-assay configuration was carried out for the detection of AFP. The excess MNPs were removed by flushing the PBST solution after the complete binding of target AFP to capture antibodies. The magneto-transport measurements were shown in Figure 10, where the resistance change of the MTJ sensor is proportional to the concentration of target AFP. As shown in Figure 10, the maximum resistance change appears at approximately 40 Oe, and the resistance changes of MTJ sensors are 17, 143, and 271 ohms for the target AFP concentrations of 0.002, 0.010, and 0.05 mg/mL, respectively. Because the concentration of MNPs binding to AFP capture antibodies is the same in the three testing groups, the resistance change is only related to the number of binding sites between the AFP antibodies and antigens. This result successfully demonstrated the feasibility of detecting AFP and thus liver cancer cell immunoassay with MNPs through MgO-based MTJs.

Mu et al. reported the detection of ricin based on TMR biosensors combining with a magnetic immune-chromatographic test strip in 2019 [12]. The TMR biosensors have a core structure of Si/SiO_2_ substrate/buffer layer/PtMn/CoFe/CoFeB (3)/MgO (2)/CoFeB (3)/NiFe/IrMn/Capping layer (nm) prepared by high vacuum magnetron sputtering. The carboxyl-modified MNPs were activated by EDC-NHS to form an NHS-active ester group. The anti-ricin antibodies are conjugated onto the surface of MNPs. The schematic structure of the magnetic immune-chromatographic test strip is shown in Figure 11, and it was placed under the TMR sensing chip. During detection, the ricin sample solutions of different concentrations ranging from 1 ng/mL to 20 μg/mL were added to the sample hole of the test strip, respectively, and then the membranes were observed to find out whether the color stripes appear on the test line and control line. After that, the test line output voltage and control line output voltage were read as V_T_ and V_C_. Next, the ricin sample solution was replaced by PBS buffer to wash the MNPs and the output voltages of both lines were read as background voltage V_0_. The voltage change on both lines was calculated as ΔV_T_ and ΔV_C_ and the ricin concentration was quantitatively determined as shown in Figure 12. It can be seen from the curves that the output voltage of each response is relatively stable and increase with the concentration of ricin solution, indicating the quantitative detection of ricin.

## 5. Magnetoresistive Biosensors for Genotyping Applications

### 5.1. AMR-Based Sensors for Genotyping

DNA identification with the help of MNPs has been extremely popular over the past decade owing to the ease of sample preparation and high sensitivity, and the integration of MR biochips into portable, handheld systems offers low-power, rapid, accurate, and real-time genotyping. The very first works on genotyping applications using ring-shaped AMR sensors were done on micron-sized magnetic beads by Miller et al. [97] in the year of 2002. As groundbreaking as this work was because it initiated the future of MR-based biochips in DNA mapping, the size requirements for the MNP tags were too large compared to the biological molecules that are seldom in the size range of micrometers. But then again, AMR chips were not that sensitive enough to detect the magnetic moments of MNPs with core sizes in the order of a few nanometers. As the quest for linear sensitive AMR sensors progressed steadily over the years [115,116,117,118], a highly sensitive AMR sensor that could detect nanometer-sized MNP at the single molecular level was in high demand. In 2016, Hien et al. [119] reported AMR-based genotyping by using a disposable card (SPA) and magnetically labeled DNA. The steps that they followed for fabrication of the SPA are given in Figure 13a. In step 1, they spin-coated a thin layer of polydimethylsiloxane (PDMS) on the silicon substrate which was then treated by ultraviolet/ozone (UVO), grafted by 3-aminopropyltriethoxy (APTES), and functionalized using succinic acid anhydride (SAA). In step 2, magnetically labeled DNA was immobilized on the SPA card. The DNA was magnetically labeled by 1 µm superparamagnetic bead, Dynabeads^®^ MyOne™ Streptavidin C1 (with 26% ferrites). The AMR sensors were fabricated in a Wheatstone Bridge configuration by 5 µm thick Ni_80_Fe_20_ material on SiO_2_/Si film using magnetron sputtering and lithography. The streptavidin-coated DNA was incubated on the SPA card which was placed at 10 µm from the AMR sensor surface. A magnetic field of 30 Oe was generated from a permanent magnet the magnetized magnetic beads enabled the detection by AMR. The authors reported the minimum detectable mass of magnetic particles to be 312 ng ferrites (approximately), which gives a detection limit of 3.8 µemu for the AMR sensors. The relationship between the mass of the target DNA and the AMR voltage is also found to be linear as shown in Figure 13b. The drawbacks of this work [119] were that despite that the Wheatstone Bridge configuration of the AMR sensors with a linear M-H loop was achieved, the size requirement for the MNPs continued to be of the order of a few micrometers. Moreover, the reported concentration requirement for the AMR-based detection of the samples was 30 µg, which is significantly high and can never pave the way for single molecular detection thereby compromising the sensitivity. Very recently, Quynh et al. [120] reported that the field sensitivity of AMR sensors can be significantly improved up to 0.56 µemu by exploiting the shape anisotropy of the single layer of 1 µm thick Ni_80_Fe_20_. This work involved a very attractive approach that included improving the sensitivity, reducing the magnetic coercive field, and reducing the thermal noise by arranging the fabricated AMR Wheatstone Bridge sensors in a series-parallel combination (see Figure 13c and Table I of Ref. [120]). The sensitivity of the series-parallel design was 1.72 times greater than the series design alone (see Figure 13d) and it could detect Fe_3_O_4_ MNPs with a core size of 50 nm. This seemed to be a significant improvement in terms of AMR-based genotyping devices from a biological point of view as it enabled the detection of nanometer-sized MNPs. In general, linear AMR-based genotyping biochips are extremely easy to fabricate but their sensitivity is compromised by the requirement of a large sample volume and large size magnetic tags.

### 5.2. GMR-Based Sensors for Genotyping

The first GMR biosensor for DNA detection was proposed in 2004 where the probe DNA-analyte DNA-MNP complex was immobilized on a sensor surface. The LOD of the sensor was reported to be 16 pg/µL [121]. The sensitivity of the GMR-based DNA detection was proven to be higher than the standard fluorescent DNA detection methods. Xu et al. demonstrated the feasibility of detecting the Polymerase Chain Reaction (PCR) products amplified from Human Papillomavirus (HPV) plasmids for the first time with pM LOD [94]. To further reduce the assay time, a one-step assay was developed [122]. The DNA product was performed by PCR on genomic DNA using a 5′-biotin forward primer and a 5′-fluorescein reverse primer. The product was then mixed with streptavidin-coated MNPs and the mixture was added to the sensor surface to bind with anti-fluorescein antibodies. The assay time was thus reduced to less than 3 min with a dynamic range of 4 to 250 pM. There have also been many efforts on integrating the amplification and denaturation processes with the GMR biosensors [123,124]. Like the immunoassay-based GMR sensing platforms, there is also a need for pushing GMR-based genotyping toward POC applications, which requires shorter assay time, improved sensitivity, and broader dynamic range. Recently, Ravi et al. reported that their GMR sensors were capable of simultaneously detecting multiple transcripts with a dynamic range of 4 orders of magnitude and a LOD of 1 pM and 0.1 pM, respectively, for 15- and 18-cycle amplified products, which demonstrated GMR biosensors’ potential to be rapid and sensitive POC platforms for genotyping [125].

### 5.3. TMR-Based Sensors for Genotyping

A proof-of-concept detection of DNA via MTJ sensors was demonstrated in 2005, where tunnel junctions with an Al_2_O_3_ barrier layer were employed to detect single-stranded DNA tagged with MnFe_2_O_4_ nanoparticles [126]. A change in the R-H curve of the MTJ sensors was observed, but no quantification of the correspondence between the concentration of DNA and the sensor signal was given. The first quantitative detection of DNA was achieved in MgO-based MTJ sensors. A total of 64 MTJ sensors were fabricated in a Wheatstone Bridge configuration, with 16 sensors connected in series within each arm [127]. The surface functionalization and the DNA assays were identical to those employed in GMR genotyping applications. After integrating with a microfluidic channel, the real-time sensor signal was obtained for 2.5 µM of the target DNA. The sensor signal first increased as the binding process took place, and a decrease in sensor signal was observed when a washing procedure was conducted. More recent research also demonstrated the detection of different concentrations of DNA extracted from *Listeria* bacterium with a sensitivity below the nM range [128]. The possibility of multiplexed assay development was also evaluated by examining the orthogonality of the DNA probes.

A comparison between different MR sensors based on magnetic tags is shown in Table 1.

## 6. MR Biosensors for Brain Mapping

Magnetoencephalography (MEG) is a technique that involves mapping the brain by recording the magnetic fields produced by the ionic currents using extremely sensitive magnetometers. Neurons are the building units of the brain and, due to the intra- and extra-cellular ionic exchange, the neuron membranes are constantly firing. This firing of neurons causes an ionic current to flow through the brain and according to the Biot-Savart Law, this time-varying current generates a circular magnetic field (see Figure 14a). Since the interconnected neurons in the brain can be considered as an infinitely long conductor, the magnetic field generated can be expressed by, $B=\frac{\mu_{0}I}{2\pi r}$, where *µ*_0_ is the magnetic permeability of free space, *I* is the current flowing through the neurons, and *r* is the distance from which the magnetic field is being measured. For a single neuron, the magnetic field generated is in the order of several pT, which attenuates gradually along the skull. Highly sensitive magnetic sensors or neural probes are required to detect this signal.

As early as 1985, Roth and Wikswo [130] made the first attempt to record the magnetic signals from the medial giant axon crayfish with the help of a toroidal pick-up coil (see Figure 14b). Since then, several systems such as giant SQUID systems [131,132] have been used to conduct magnetic recordings from the brain (see Figure 14c). But all of them are huge instruments that are only able to record the whole brain and significantly lack the focal, localized precision like neural probes. Aside from magnetic signal recording, there are several other available forms of neural recording systems such as the 3D Utah arrays [133], the Michigan arrays [134], electroencephalogram (EEG), magnetic resonance imaging (MRI), computed tomography (CT) scan, positron emission tomography (PET) scan, etc. However, each technique has its own pros and cons. The existing neural probes that record the electrical signal from the neurons are in galvanic contact with the neural tissue and hence the electrochemical exchange between the probes and the tissues causes glial cell scarring around the probes, which, as a result, causes an increase in the impedance of the system. Although MEG can measure brain activities very quickly, the MEG scanner systems are bulky, involving a time-consuming procedure. Furthermore, MEG systems cannot measure brain activities from a focused region of the brain, which MRI scanning techniques or implantable recording electrodes can offer [135,136]. EEG signals suffer from heavy attenuation along the skull and hence the amplitude of the measured signal is compromised [137]. With respect to structural scanning techniques, CT scan has high spatial resolution but shows limited contrast with tissues. MRI systems show medium spatial resolution, low temporal resolution, and are extremely expensive to run. Since the permeability of the biological tissues is like the permeability of free space, the magnetic field recordings from the brain have high spatial as well as temporal resolution. In this respect, linear MR sensors [138,139,140,141,142] have been investigated for neural signal recording.

The first-ever attempt to record neural activity by MR sensors was made by Amaral et al. [138] in the year of 2011 and since then there have been some relevant follow-up works. The SV-GMR sensor stack and the in vitro experimental set-up used to record the brain activity is shown in Figure 15a,b. The neural response due to firing of 500 neurons as recorded by the GMR sensor array is shown in Figure 15c. The neural response recorded by the GMR sensors had an amplitude of 500 µV and lasted for 20–30 ms. The noise level of the GMR sensor for 1 mA current biasing was 1 µV for 70 Hz bandwidth and the field noise was found to be 125 nT. The absence of any signal on the addition of the tetrodotoxin (TTX) drug that supposedly blocks neural activity (see Figure 15d) validates the fact that the signals picked up by the GMR sensors can be attributed to some form of neural response.

Later in 2017, Caruso et al. [140] used five segments of a 4 × 30 µm^2^ SV-GMR needle to study in vivo neural response from a cat’s visual cortex (see Figure 16b). Although their approach was interesting, the neural response recorded is still obscure. At about the same time, a similar platform for in vitro magnetic neural recording from an array of 12 MTJs, each of 3 × 40 µm^2^, was reported by Sharma et al. [141]. According to their report, for a bias field of 6 mT for noise measurements, the integrated noise observed was at 250 nT (RMS) for a frequency range of 1 Hz to 1 kHz. Their work validated the MTJ neural signal recording with that of the standard local field potential (LFP) response and the MTJ recording seems to resemble well with the LFP recordings (see Figure 16a). In 2018, a Wheatstone Bridge configuration of the TMR sensors was employed to study the neural response. Fujiwara et al. [142] reported that those TMR sensors in the Wheatstone Bridge arrangement were successful in mapping the neuron response. Figure 16c was their reported neural response.

MTJ sensors are composed of elements such as Co, Fe, Mn, etc., which are extremely toxic to the human brain. Therefore, the investigation of the MR sensor-based neural probes requires an additional step, which is making the neural probes biocompatible. Moretti et al. [67] reported that the cells are viable on CoFeB/MgO/CoFeB-based MTJ sensors for up to 31 days in vitro when the MTJ sensors were passivated by physical vapor deposited (PVD) trilayer of SiO_2_(50)/Si_3_N_4_(25)/SiO_2_(50) (in nm); the SiO_2_ layers were deposited by e-beam evaporation and the Si_3_N_4_ layer were deposited by magnetron sputtering. Interestingly, this work [67] reported that chemical vapor deposition (CVD) of the trilayer of the same thickness resulted in a lack of cell viability. This unexplained phenomenon can be attributed to the fact that CVD involves the use of toxic precursors (SiCl_4_), which might remain in the thin film after the deposition thereby resulting in cell death.

Another important aspect relating to neuron signal sensing using MR sensors is the orientation of these sensors with respect to the neurons. As obtained from Figure 13a, a circular magnetic field (B) is generated on applying the Biot-Savart Law to ionic neuronal currents. Not only is it necessary for the neurons to locate on top of the MR sensor pillars, but it is also important for the neurons to lie along the appropriate sensing direction of the MR sensors to allow maximum sensitivity. Depending on the amount of magnetic field detected, the output amplitude of the MR signal will vary. For all the reported works on neuron recordings using MR sensors so far, the orientation dependence of the MR sensor signal for detecting most of the nT-range magnetic field from the neurons has not been investigated experimentally. It is understandable that the neuronal circuitry in the brain is quite random and there is no good way to judge the orientation of neurons with respect to the MR sensors. However, the best efforts were made by Moretti et al. [67] where they monitored the in vitro growth of neurons directly on the MTJ pillars to minimize the distance between the MR sensors and the neurons. In addition, they tried to orient the axons perpendicular to the MR sensing direction (see Figure 15d), such that the circular magnetic field can be detected along those axes. As the neurons in different areas of the brain are variedly oriented, one must customize the MR sensor patterns and define the sensing direction of the MR sensors such that the axons along that region of the brain lie perpendicularly oriented when the MR sensor probe is inserted. Future study is needed to further address the orientation dependence of MR sensor signal in neuron recording applications.

Nevertheless, neural recording with MR biosensors still has a long way to go. The ultra-low and slow signal is hard to detect via MR sensors as the noise in the low-frequency regime is currently comparable and even higher than the signal. This low-frequency noise (1/f noise), defined by phenomenological models [53,144], lacks proper ways to be reduced or eliminated. Although Amaral et al.’s [138] work with GMR as a neural recording probe in an in vitro set-up showed promising results, it lacks any experimental corroboration from other existing techniques of recording a neural response, e.g., LFP recording or patch-clamp recording. Sharma et al.’s [141] work on in vitro magnetic recording using MTJ sensor array is also promising but their experiments lack an adequate number of repeats and controls. Despite attempts to study the in vivo magnetic responses of SV-GMR sensors, the reports published require further validation. Most importantly, the research in the field of magnetic neural response recording lacks a standard operating procedure that could define the sensor operation and how it picks up the neural signals. No theoretical investigation has been carried out to study the sensitivity of the sensor required to detect the neural response. Further research on brain mapping using MR biosensors is in urgent need of this study.

## 7. Flexible Magnetoresistive Biosensors

Flexible electronics have been of great interest in the past years, especially for biological applications due to their ability to adapt to different shapes of biological surfaces. Compared to other types of sensors, the capability of responding to an external magnetic field makes flexible magnetic biosensors potential candidates for body tracking, evaluating the effectiveness of drug delivery, disease monitoring, etc. There are two most used techniques to fabricate flexible MR devices. One is to directly deposit MR stacks on flexible substrates [145] and the other one is to deposit the MR stacks on silicon substrates first, and then transfer the MR stacks to flexible substrates either by adding a sacrificial layer between the silicon substrates and the MR stacks or by etching away the silicon substrate after the deposition of the stacks [146,147]. There have been some very comprehensive reviews of flexible magnetic devices [148,149]. Here, we will only focus on a novel fabrication technique called printed MR devices and the fabrication of flexible MTJ devices as they are the most difficult MR devices to fabricate compared to AMR and GMR devices due to more complexed fabrication processes and higher requirements to film quality and surface roughness.

### 7.1. Printable MR Devices

Besides directly depositing MR stacks onto flexible substrates and transferring the patterned MR devices from rigid substrates to flexible substrates, the printing of the “magnetic ink” onto flexible substrates is another promising technique toward rapid and low-cost flexible MR device fabrication. The “magnetic ink” containing flakes of MR stack composites is dispersed in binder solutions. To fabricate the MR flakes, MR thin films are firstly deposited onto a silicon wafer coated with photoresist. The films are then released from the substrate via a lift-off process, which shows flake-like structures due to the intrinsic strain. A ball milling process is then performed and filtered to obtain flakes with desired sizes. Finally, the flakes are re-dispersed into binder solutions to get the magnetoresistive ink (Figure 17). For GMR multilayer flakes, the resulting MR value varies from 4.5% on paper to 8% on rigid silicon wafers [150]. It has been shown that the MR, sensitivity, and the operating temperature range of the printed MR sensors could be further improved by reducing the size of the flakes and altering the polymeric binder solution. The maximum MR achieved in multilayer GMR flakes was 37% with a sensitivity of 0.93 T^−1^ at 130 mT, which was comparable to the state-of-art GMR sensors [151].

### 7.2. Flexible MTJs

As the flexibility relies mostly on the substrate, looking for a compatible substrate with the flexibility and capability of growing a good magnetic thin film becomes essential. In 2010, the first flexible MTJ device was developed with Co/Al_2_O_3_/Co stacks grown on a transparent flexible organic substrate called “gel film” [152]. To reduce the surface roughness of the flexible substrate, two successive layers of photoresist were spin-coated on the substrate, followed by the deposition of an organic butter layer. A TMR ratio of 12.5% was preserved after mechanical bending, which demonstrated the feasibility of fabricating flexible MTJ devices for the first time. Later, Amilcar et al. successfully deposited Co/Al_2_O_3_/NiFe MTJ on Kapton [153]. This device achieved a bending radius to 5 mm without any degradation of the TMR ratio

To develop MTJ devices with a higher TMR ratio, people have turned to MgO-barrier MTJs. Comparing to AlOx-barrier MTJs, MgO-barrier MTJs have significantly higher TMR but are also more demanding on the thin film preparation process. Firstly, they require even lower surface roughness than the AlOx-based MTJs, which organic substrates could hardly satisfy. Secondly, most of the organic substrates cannot survive the high annealing temperature (usually higher than 300 °C). Thirdly, few organic substrates could withstand microfabrication processes such as etching and developing. Two methods have been brought up to solve these three major challenges during the development of MgO-barrier MTJs with a high TMR ratio.

The first method was brought up by Loong et al. [154] in 2016 along with the first flexible MgO-barrier MTJ. They proposed a “transfer print process”, which was to deposit the magnetic thin films on the traditional silicon substrate, followed by the etching of the silicon substrate to release the MTJs. Then, the MTJ was transferred to a flexible substrate such as PET and PDMS. The TMR ratio after the transfer print process not only preserved but also got improved by 0.38 times than the original one. Another method was proposed by Chen et al. [147] in 2017, where flexible MgO-barrier MTJs were developed by etching the silicon substrate down to 14 µm. The MTJ was firstly prepared through the conventional process and then a layer of photoresist was spin-coated onto the surface. Next, the chip was turned over and mounted onto a 4-inch wafer and the silicon substrate on the backside was etched by deep trench etcher system. The silicon on the backside was etched away with SF_6_ and argon plasma until the device became flexible at an ultra-low thickness. The device could preserve a 190% TMR ratio up to 500 times bending cycles with a bending radius of 15 mm.

## 8. Conclusions and Outlook

The ability to convert magnetic signals to electric signals inspires tremendous research interests in MR sensors. Due to their low background noise, compatibility with nanofabrication technology, and capability of multiplexed detection, much progress has been made toward detecting biological signals with MR sensors. Over the past 20 years, magnetic biosensors based on AMR, GMR, and TMR effects have all been investigated. Compared to GMR and TMR biosensors, the operating field of AMR biosensors is smaller. However, the low MR ratio and fragility at high temperatures largely limit the applications of AMR biosensors. Of all three types of MR biosensors, GMR biosensors possess many advantages such as moderate MR ratio, simplicity in nanofabrication process, and high linearity, and thus is the most widely employed technologies in the literature. Although TMR biosensors exhibit the highest MR value, their large noise and complicated fabrication process, and the need for top electrodes, are the shortcomings with respect to biomarker detection. However, TMR sensors still have a large potential in the detection of ultra-weak biomagnetic field if their noise level is well controlled.

Recent advances in MR biosensors have been focusing on employing different technologies to achieve highly sensitive, biocompatible, and rapid sensing performance. For the direct detection of biomagnetic field, the sensitivity of the MR sensors becomes important since the magnetic field from the human body can be as low as several pT. The optimization of the MR stack structure is the most straightforward way to increase the sensitivity of the MR sensors while maintaining moderate linearity. For example, several new TMR structures have been proposed for sensing applications such as the superparamagnetic free layer [155] and two pinned antiferromagnetic electrodes [156]. Other sensitivity improvement from the circuitry level have also been employed such as integrating multiple MR sensors into a Wheatstone Bridge [157], connecting multiple MR sensors in series [158], and the application of MFCs. For the in vitro biological detection based on magnetic tags, the efficiency of capturing magnetic tags on the sensor surface is another crucial factor. Since the sensor signal has been proved to be highly dependent on the position of the magnetic tags on the sensing element due to the influence of the stray field from the edge of the MR stacks, the geometry of the MR biosensors needs to be carefully designed to optimize the sensitivity as well as the uniformity of the sensor signal. Besides the lateral distribution of the magnetic tags, the signal dependence on the vertical distance between magnetic tags and the sensor signal has also been investigated, which can be adjusted by the passivation layer thickness and the structure of the magnetic bioassays. The integration of MR biosensors with microfluidic channels also reduces the distance between the magnetic tags in the solution and the sensor surface, thus decreasing the assay preparation time and increasing the sensitivity of the magnetic bioassays.

New MR biosensing platforms such as the POC devices as well as the integration with flexible substrates have also been developed by multiple groups. For POC devices, the increased noise level and the requirement for reduced assay preparation time are the key problems. Several POC devices are proposed with optimized PCB board configurations as well as strategies to reduce the assay time such as the employment of microfluidic channels and wash-free bioassays. For the integration with flexible substrates, the major challenge comes from the engineering of surface roughness during the deposition of MR stacks on polymeric substrates as well as the influence of induced strain in the bending process on the performance of the flexible MR sensors. A cost-effective way to fabricate flexible MR stacks in large-scale is also a future direction in the field of flexible MR biosensors.

## Figures and Tables

**Figure 1 micromachines-11-00034-f001:**
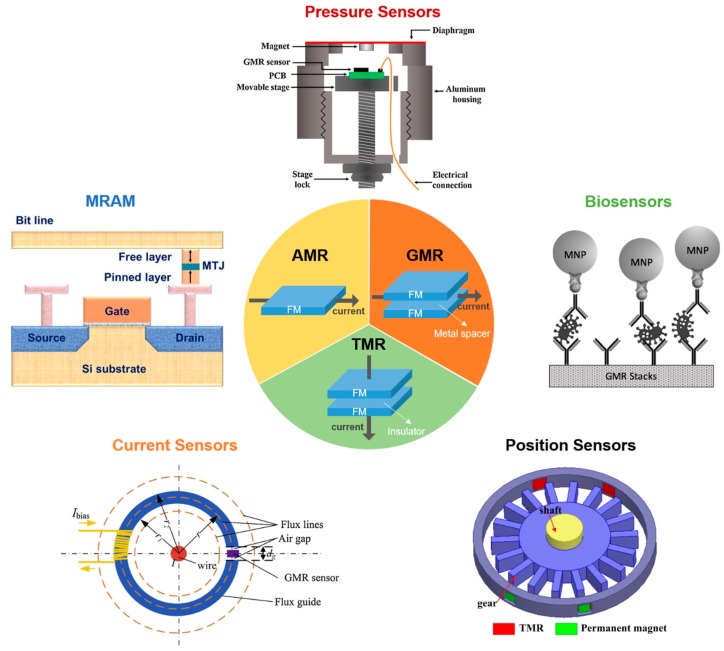
Schematic illustration of different types of magnetoresistance (MR) sensors and their applications. Reprinted from Ref. [3,5,22,23] with the permission from Elsevier 2019, 2017, 2018, and 2018, respectively.

**Figure 2 micromachines-11-00034-f002:**
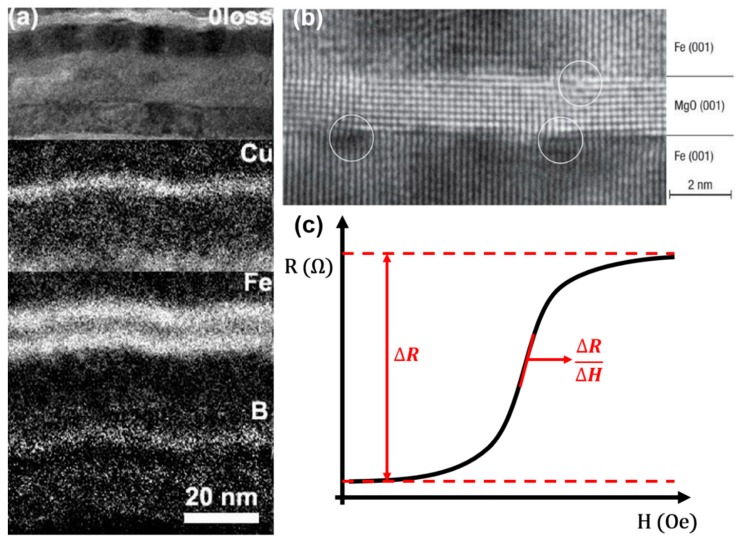
(**a**) Energy filtered TEM image of a giant magnetoresistance (GMR) structure of oFe (1.5 nm)/Cu (50 nm)/IrMn (10 nm)/CoFeB (6 nm)/Cu (2.5 nm)/CoFeB (6 nm)/Ru (8 nm) [51]; (**b**) MgO-based magnetic tunnel junction (MTJ) with 180% tunneling magnetoresistance (TMR) ratio reported by Yuasa et al. [52]; (**c**) a typical transfer curve of the magnetoresistive sensors. Reprinted with permission from AIP Publishing 2010 (**a**) and Springer Nature 2004 (**b**).

**Figure 3 micromachines-11-00034-f003:**
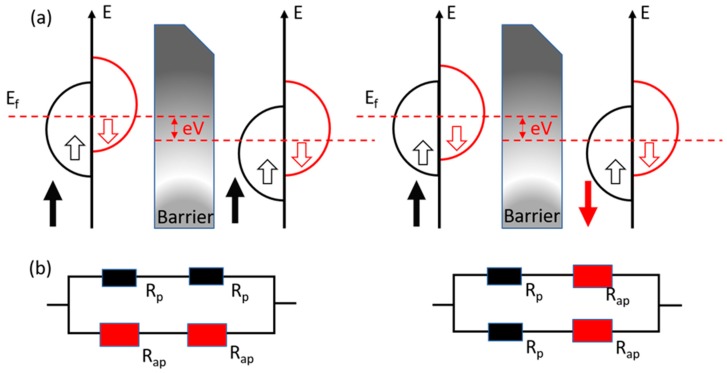
(**a**) Schematic of density of electronic states with parallel magnetization (PM) and antiparallel magnetization (APM). E, the electron energy; E_F_, the Fermi level; (**b**) Two-current model of tunneling magnetoresistance.

**Figure 4 micromachines-11-00034-f004:**
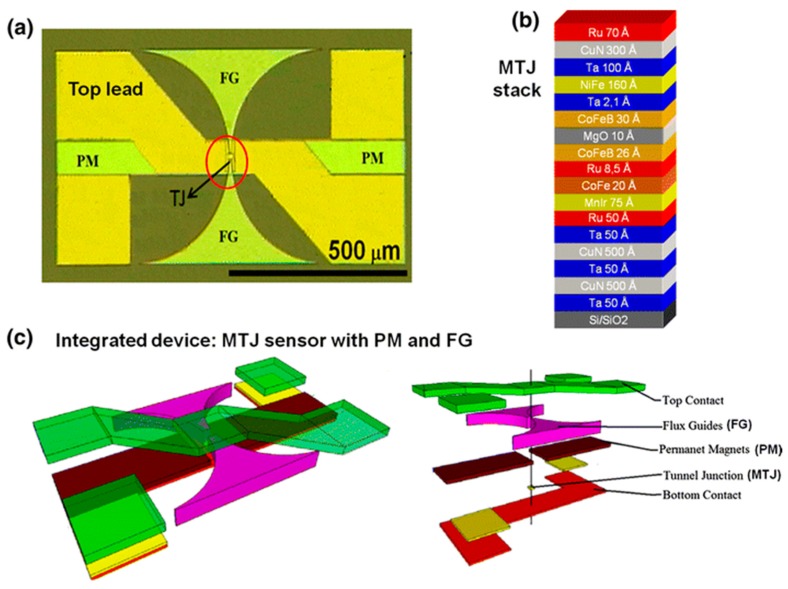
(**a**) Top view picture of one MTJ integrated with permanent magnets and MFCs; (**b**) stack structure of the MTJ devices; (**c**) final layout of the devices [71]. Reprinted with permission from Springer-Verlag Berlin Heidelberg 2014.

**Figure 5 micromachines-11-00034-f005:**
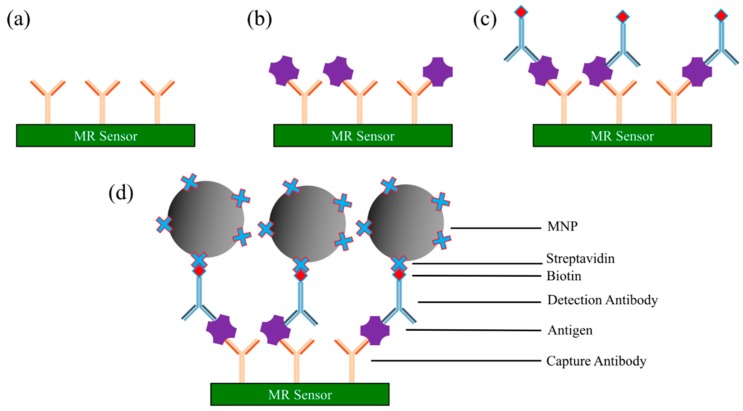
Schematic view of a MR sensor-based sandwich assay. (**a**) MR sensor surface coated with capture antibodies; (**b**) biological sample is loaded onto sensor surface, and antigens of interest specifically bind to capture antibodies; (**c**) biotinylated detection antibodies are added and specifically bind to antigens; (**d**) streptavidin-linked magnetic nanoparticles (MNPs) are added and bind to detection antibodies through biotin-streptavidin conjugation.

**Figure 6 micromachines-11-00034-f006:**
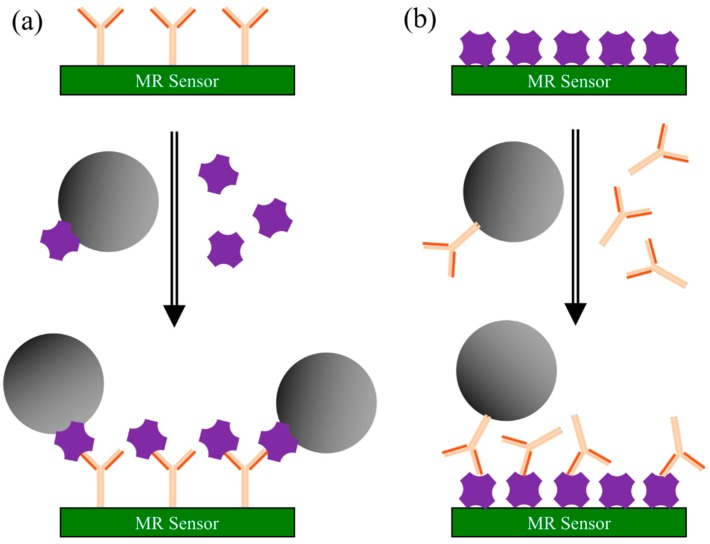
Schematic view of competitive binding assays. (**a**) MR sensor surface pre-coated with capture antibodies. The MNP-labeled antigens are competing with unlabeled antigens from testing samples for the antibody binding sites; (**b**) MR sensor surface pre-coated with antigens. The MNP-labeled antibodies are competing with unlabeled antibodies from testing samples for the antigen binding sites.

**Figure 7 micromachines-11-00034-f007:**
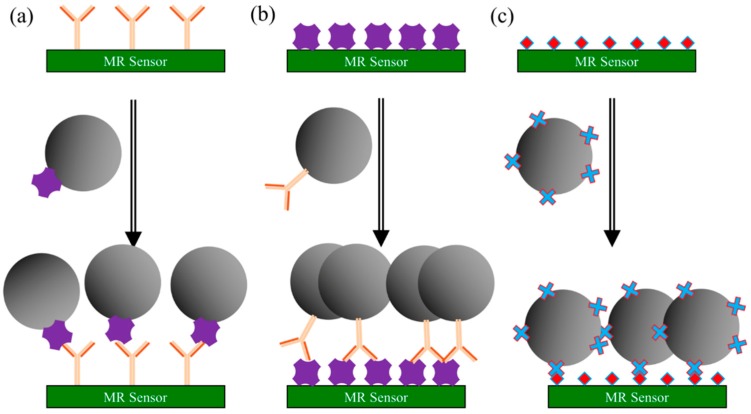
Schematic view of direct assays. (**a**) MR sensor surface pre-coated with target antibodies from testing sample followed by adding MNP-linked antigens for specific binding. The number of captured MNPs through antigen-antibody conjugations is proportional to the electrical/magnetic signals from MR sensors; (**b**) MR sensor surface pre-coated with target antigens from testing sample followed by adding MNP-linked antibodies; (**c**) MR sensor surface pre-coated with biotin followed by adding streptavidin-coated MNPs. This structure is mostly used as a positive control.

**Figure 8 micromachines-11-00034-f008:**
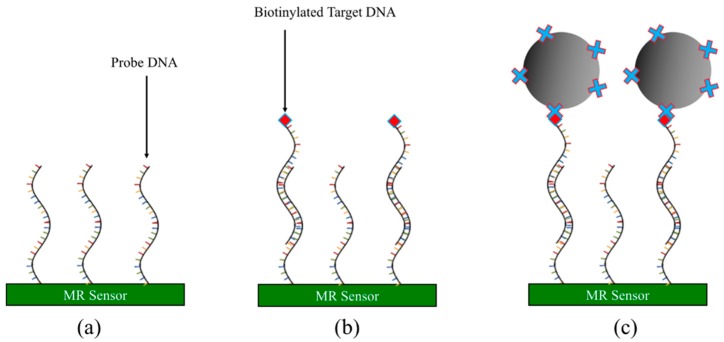
Schematic view of DNA-based assay. (**a**) Probe DNA strands immobilized onto MR sensor surface; (**b**) biotinylated target DNA hybridizes with probe DNA; (**c**) streptavidin-coated MNPs captured by target DNAs through biotin-streptavidin conjugation.

**Figure 9 micromachines-11-00034-f009:**
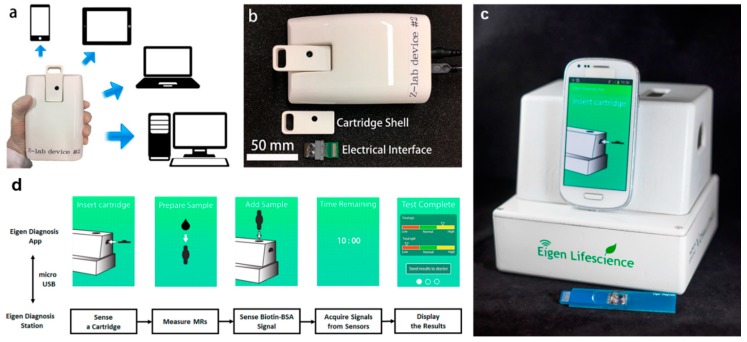
(**a**) The GMR handheld system can communicate with cell phones, laptops, tablets, and computers from Ref. [64]; (**b**) layout of the handheld system with the cartridge shell and electrical interface in Ref. [64]; (**c**) the Eigen Diagnosis Platforms (EDP) system from Ref. [20]; (**d**) the block diagram of the test process of the EDP system in Ref. [20]. The straightforward interface and process can lead users through the testing process and the results can be shown 10 min after the sample addition. Reprinted with permission from American Chemical Society 2017.

**Figure 10 micromachines-11-00034-f010:**
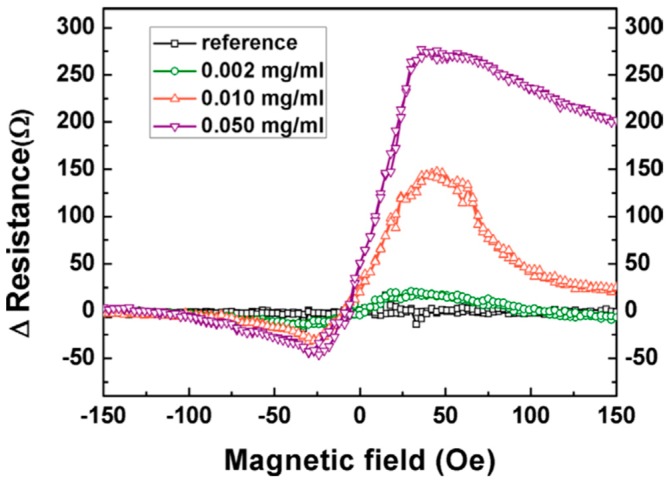
Resistance change of MTJ sensor after binding with three different concentrations of alpha-fetoprotein (AFP) antigens [114]. Reprinted with permission from AIP Publishing 2012.

**Figure 11 micromachines-11-00034-f011:**
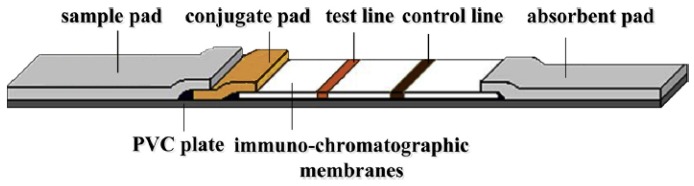
Schematic structure of the magnetic immuno-chromatographic test strip [12]. Reprinted with permission from Elsevier 2019.

**Figure 12 micromachines-11-00034-f012:**
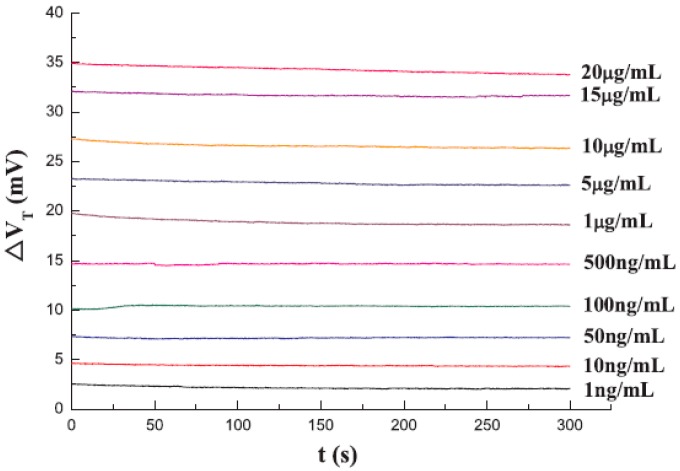
TMR biosensor signal response to the aggregated magnetic particles on T line with different concentrations of ricin [12]. Reprinted with permission from Elsevier 2019.

**Figure 13 micromachines-11-00034-f013:**
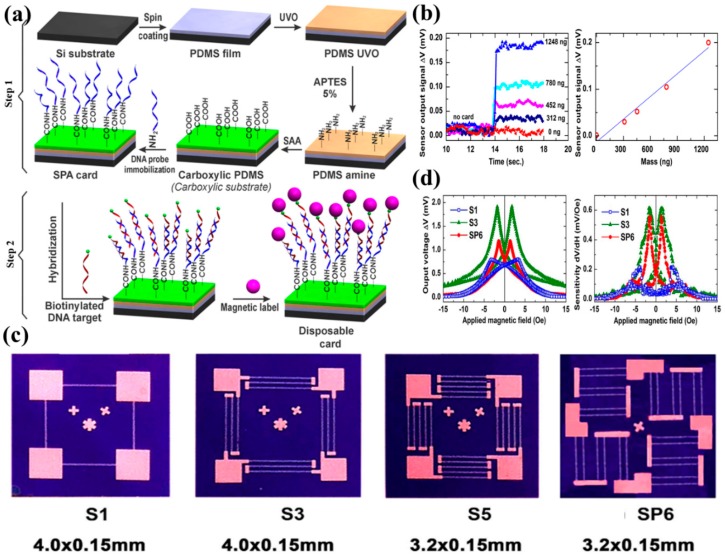
(**a**) The fabrication steps of the disposable SPA card and immobilization of DNA with magnetic labels as the biomarkers [119]; (**b**) the AMR sensor output with varying concentration of the MNPs and representing the lowest detectable amount of MNPs is 312 ng [119]; (**c**) the optical microscopic image of the fabricated Wheatstone Bridge AMR sensors with a single resistor (S1), three resistors in series (S3), five resistors in series (S5), and six resistors in series and parallel configuration (SP6) [120]; (**d**) magnetoresistance (in mV) and sensitivity (in mV/Oe) dependence on magnetic field for the designed S1, S3, and SP6 patterns [120]. (**a**,**b**) Reprinted under the terms of the Creative Commons Attribution 3.0 license. (**c**,**d**) Reprinted with permission from The Minerals, Metals & Materials Society 2018.

**Figure 14 micromachines-11-00034-f014:**
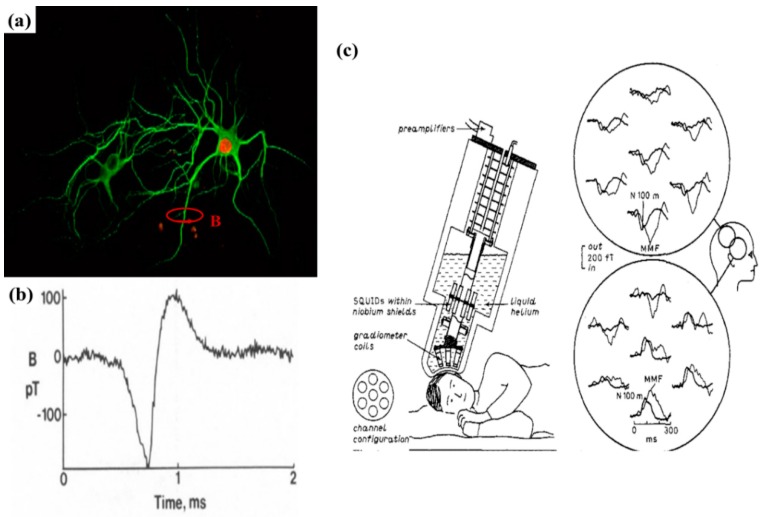
(**a**) The firing of neurons causes an ionic current to flow which according to the Biot-Savart Law generates a magnetic field (B) around the neurons that can be measured by magnetic sensors [143]; (**b**) the first recording of the magnetic field picked up from a single neuron with the help of a toroidal pick-up coil as reported by Roth and Wikswo [130]; (**c**) Superconducting Quantum Interference Devices (SQUID) system set up to record magnetic field from the cerebral cortex of the human brain. The circular insets show the real-time magnetic field recording from the brain [132]. (**a**) Reprinted under the terms of the Creative Commons Attribution 4.0 International License (CC BY). (**b**) Reprinted with permission from Elsevier 1985. (**c**) Reprinted with permission from IOP Publishing 1989.

**Figure 15 micromachines-11-00034-f015:**
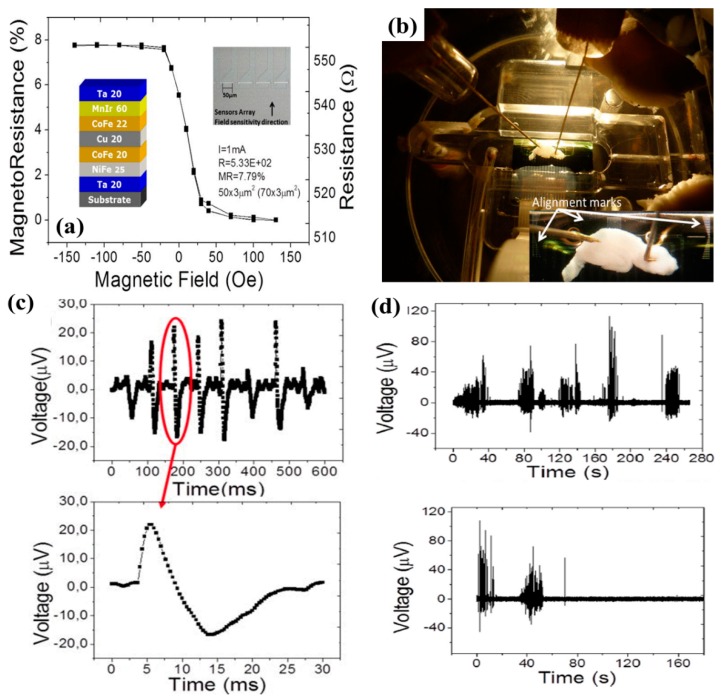
(**a**) The SV-GMR sensor stack arranged in an array of 15 sensors, each of 3 × 15 µm^2^ in cross-section. The R-H curves shows that it is a linear sensor having an MR ratio of 7.8% and a sensitivity of 0.15%/Oe [138]; (**b**) in vitro system set-up for recording neural response from mice hippocampal slices [139]; (**c**) the real-time neural response as recorded from the SV-GMR sensor array [138]; (**d**) GMR sensor recording from the hippocampal slices with (bottom) and without (top) the addition of the tetrodotoxin (TTX) drug, which blocks neural response [138]. Reprinted with permission from AIP Publishing 2011.

**Figure 16 micromachines-11-00034-f016:**
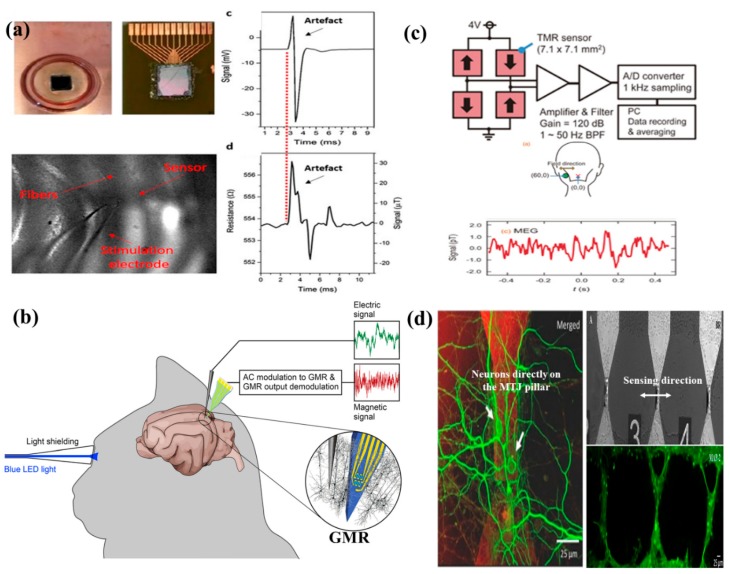
(**a**) MTJ sensor array and the in vitro experimental set-up demonstrating the neural response. The signal (in mV) and the resistance (in Ω) are the local field potential (LFP) recordings and MTJ signal response, respectively. The shape of both the signal resembles each other [141]; (**b**) in vivo experimental set-up to record neural response from rat cerebral cortex [140]; (**c**) Four TMR sensors arranged in Wheatstone Bridge configuration along with neural response as recorded by the sensor configuration (in pT) [142]. (**a**) Reprinted under a Creative Commons Attribution (CC BY) license. (**b**) Reprinted with permission from Elsevier 2017. (**c**) Reprinted under the terms of the Creative Commons Attribution 4.0 license. (**d**) Orientation of the axons of the neurons perpendicular to the sensing direction of the MTJ pillars to facilitate most optimized detection of the magnetic field generated by the ionic currents flowing through the axons [67].

**Figure 17 micromachines-11-00034-f017:**
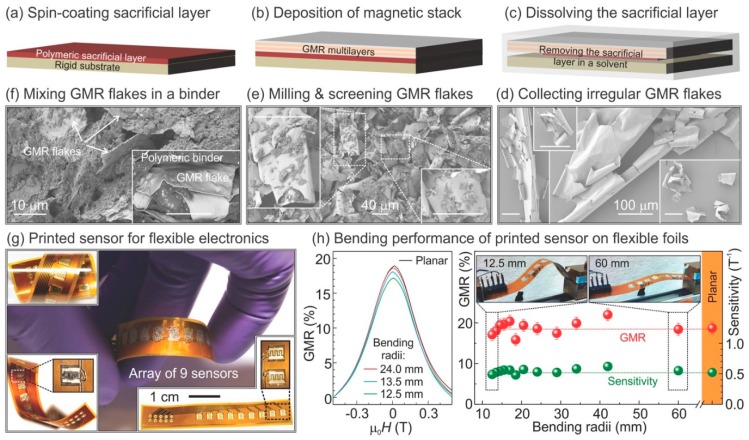
(**a**–**f**) Schematic illustration of the fabrication process of the GMR flakes; (**g**) printed GMR sensor arrays on flexible substrate; (**h**) evolution of magnetoelectrical performance of the printed GMR sensors upon different bending radius [151]. Reprinted under the terms of the Creative Commons Attribution Non-Commercial License.

**Table 1 micromachines-11-00034-t001:** Comparisons between different magnetoresistive biosensors.

MR Sensor	Magnetic Label	Assay Time	Target Analyte	Limit of Detection (LoD)	Matrices	References
AMR	50 nm Fe_3_O_4_	NA	Fe_3_O_4_ MNP	0.56 µemu	NA	[120]
AMR	1 μm Fe_3_O_4_	3 h	ssDNA	4.5 pM	HCl, EDTA, NaCl buffer	[119]
AMR	4.3 μm NiFe	NA	NiFe Microbead	Single particle	NA	[97]
GMR	50 nm Fe_3_O_4_	NA	ssDNA	0.1 pM	Saline sodium citrate	[125]
GMR	50 nm Fe_3_O_4_	1 h	Influenza A virus	0.3 nM	Swine Nasal Swab Sample	[21]
GMR	50 nm Fe_3_O_4_	2 h	Ovarian cancer biomarkers	7.4 pg/mL	PBS	[18]
GMR	50 nm Fe_3_O_4_	3 min	LamB gene of E-Coli	4 pM	Borate buffer	[122]
GMR	0.35 μm Fe_3_O_4_	12 h	ssDNA	10 ng/μL	35% formamide solution	[121]
TMR	16 and 50 nm Fe_3_O_4_	2 h	ssDNA	2.5 μM	PBS	[129]
TMR	20 nm Fe_3_O_4_	NA	AFP antigens	0.002 mg/mL	PBS	[114]
TMR	16 nm Fe_3_O_4_	NA	ssDNA	100 nM	DI water	[127]

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
