# Peer review of "Advances in Magnetoresistive Biosensors"

_micromachines, 2019, doi:10.3390/mi11010034_

Round 1

Reviewer 1 Report

The paper is very well organized, it is easy to read and provides a fundamental overview of the topic of magnetoresistive biosensors, as well as the current challenges in the field

However, there are a few improvements that can be made to the paper, with emphasis on section 6.

The quality of figure 1 should be improved, since some text is almost illegible. In section 2, it would be beneficial to include figures of the stack not only for TMR (2.3) but also for the GMR (2.2), and to clarify the difference between sensitivity and magnetoresistive ratio, because high MR does not necessarily imply higher sensitivity. An easy way to do this would be to add a figure with typical transfer curves. Also, it would be interesting for the authors to comment on the signal to noise ratio for both TMRs and GMRs, given that TMRs are typically more noisy. Section 3.1.1: should include discussion regarding measurement throughput, since one important performance metric is the time required for the analyte to be measured. Figure 3 should be improved, specifically the font size. Section 4.2: in terms of multiplexed detection, this requires increase number of sensors if one ones to keep the same number of sensors/spot. Increase number of sensors leads to more complex electronics readout in POC applications, so it would be interesting to see a comment on these tradeoffs. Section 6: the magnetic fields are produced by ionic currents, not electrical currents, since the charge carrier in the neurons are not electrons but ions. This should be fixed. Section 6: It would be relevant to include a discussion on the influence of the orientation of the eithrer GMR or TMR sensors in reference to the target neurons, since it can influence the measured amplitude. Section 6: “Traditional MEG systems are bulky, and the testing requires investment of a large amount of time and money by the patients.” These sentence should either improved or removed, since it is rather vague and seems out of place. Section 6: “EEG signals suffer from heavy attenuation along the skull and hence are noisy”, this sentence should also be improved, since signal attenuation does not relate to noise, just a decrease in signal amplitude. Section 6: “CT scan has high spatial resolution but shows limited contrast with tissues”. The authors here seem to confuse neuronal signal recording with structural imaging. Section 6: ” In addition, all the existing systems for neural recording lack neural selectivity”. This sentence is absolutely not true and should be changed or removed. There are several tools and techniques that allow not only for high spatial resolution but even to single neuron resolution with maximum selectivity, such as high density MEAs, patch clamping, or calcium imaging. Section 6: Figure 13. Ionic current, not electric current Section 6: ‘”with a field noise of 6 mT was reported”. The 6 mT is not the field noise, since with this level of noise no neuronal signals would be able to be recorded. The 6 mT is the biasing field that was used for the noise measurements. The integrated noise from 1 Hz to 1 kHz was 250 nTrms. This should be corrected. Section 6: “GMR sensors can be made to operate in the linear regime with ease but its field sensitivity is compromised. MTJ sensors have high field sensitivity but has a relatively narrow linear regime of operation.” This sentence is relevant not only for neuronal recording, and should be in section 2, as well.

Reviewer 2 Report

The paper by Su et al. is a comprehensive review of the last twenty years of advances in the field of magnetoresistive biosensors. I think that the authors gave a good overview of the topic and therefore I suggest publication after the following points have been addressed.

1) Line 123: In order to be clearer in the phenomenological description of TMR, I would clarify the sentence and maybe add the schematic of the two current model for parallel and anti-parallel alignment of the magnetizations, showing the band structure of the ferromagnets in the two configurations.

2) In paragraph 3.1.1, the authors reported the conclusions of Lee et al., which suggested the MNPs on the edge of the sensor could contribute with more signal. This is true, however, in order to achieve signal quantification, the narrow signal peak at the edges should be avoided (see for instance Li et al., J. Appl. Phys. 99 (2006) 08P107 and Albisetti et al., Sensors and Actuators B 200 (2014) 39–46). The authors should discuss also this aspect.

3) Regarding the microfluidic paragraph 3.1.4, the authors could also refer to magnetic domain wall tweezer technology, which can be exploit for the manipulation of biofunctionalized MNPs (Donolato et al., Adv. Mater. 2010, 22, 2706–2710; Rapoport et al. Lab Chip. 12, 4433-4440 (2012)). Recently, integration with AMR sensors for single particles detection has been also demonstrated (see Monticelli et al., Small 12, 921-929).

4) At the end of section 6 (line 615), the author says that MTJs have relative narrow regime of linear operation respect with GMR. I am not convinced that this is the point and the main problem of MTJ regarding the detection of biological magnetic field (see for instance the characteristic reported in Shen et al., J. Appl. Phys. 103, 07A306 (2008)). On the contrary, I think that 1/f noise is the main disadvantage of MTJs for this application, due to the relative slow signal coming from the brain activity. This is reported in some sense in the conclusion, but maybe it is worth to be mentioned at the end of section 6.

5) Regarding Flexible Magnetoresistive Biosensors (section 7), the author should make a brief introduction about the possible applications of such sensors before going into details. Indeed, it seems that there are still no applications of this technology to a biological problem. If this is the case, I suggest listing the potential applications. And what about AMR on flexible substrates?

6) Paragraph 7.1: Why only with MgO-based MTJs are there some issues related to the fabrication process (etching and development as reported in line 642) and not with Al2O3 MTJs? The authors should clarify this point.

Minor points:

The author should revised the text, checking for typos and revise English. I found some errors:

The authors should carefully check all the references: some are duplicated. For instance [61] and [80] ,[63] and [140], [79] and [102], [89] and [120], are the same. Reference [116] is not pertinent with the text where it is cited (line 452). The authors are describing AMR sensors and the reference is related to GMR. I suggest to standardize the style of the references: some for instance have the title written in capital letters. It is hard to see the text in some parts of Fig. 1, I suggest to enlarge it. Line 78: I would say spin-orbit interaction and not spin-orbital Line 130: Julliere, not Michel I suggest to move paragraph 7.2 (about GMR) before paragraph 7.1 for consistency with the organization of the previous sections Line 616: their and not its. Figure 16 should be scaled to fit the paragraph and centered.

Round 2

Reviewer 1 Report

Overall, the authors did a good job in answering to the previous comments. 

Just one remark regarding comment #11.

Several works have been published where patch clamping and calcium imaging recording modalities are employed in vivo (examples below). Moreover, as the authors also referred to in the cover letter, MEAs are used for in vivo recording, and have been shown to be able to have electrode sizes with similar surface area as single neurons, thus having approximately neuron spatial selectivity. For the reasons above, saying that "all the in vivo systems for neural recording lack spatial selectivity" is a very strong statement that is not taking into account the abovementioned recording modalities, and should still be revised.

Yoshida, E., Terada, S., Tanaka, Y.H. et al. In vivo wide-field calcium imaging of mouse thalamocortical synapses with an 8 K ultra-high-definition camera. Sci Rep 8, 8324 (2018) doi:10.1038/s41598-018-26566-3

Doyun Lee and Albert K. Lee, In Vivo Patch-Clamp Recording in Awake Head-Fixed Rodents, Cold Spring Harb Protoc; 2017

Author Response

Thanks for the comment. We agree that this statement is inaccurate and have removed the sentence from the manuscript.

Reviewer 2 Report

The authors have improved the manuscript, which is now suitable for publication.

Author Response

Thanks for your effort in review this manuscript.